# Invertebrate composting quality of the invasive alga *Rugulopteryx okamurae*, prospects for its bio-recycling, management and circular economy

Daniel Patón[1]*, José Carlos García-Gómez[2]*

1 Área de Ecología, Facultad de Ciencias, Universidad de Extremadura, Badajoz, Spain, 2 Laboratorio Biología Marina, Seville Aquarium R + D + I Biological Research Area, Department of Zoology, Faculty of Biology, University of Sevilla, Sevilla, Spain / Marine Biology Station of the Strait, Ceuta, Spain

☯ These authors contributed equally to this work.
* dpaton@unex.es (DP); jcgarcia@us.es (JCGG)

## Abstract

In recent decades, the invasive seaweed *Rugulopteryx okamurae* has had a huge environmental impact on marine biodiversity, fisheries, GHG emissions and public health along much of the Iberian Peninsula and islands coastline. Due to the enormous amount of algae biomass that is expelled to the beaches where it slowly rots, some circular economy business initiatives, such as composting, are emerging. In the present study, we compared the quality of compost obtained from earthworms (*Dendrobaena veneta*), cockroaches (*Eublaberus sp.*), mealworms (*Tenebrio molitor*) and black soldier fly larvae (*Hermetia illucens*). Batches fed with 100% organic kitchen waste (control group) were compared with batches fed with 50% algae and 50% kitchen waste (treatment group). Our results show that the most sensitive species (*D. veneta* and *T. molitor*) to *R. okamurae* toxins compost adequately. The C/N ratio, electrical conductivity (EC), pH, total organic matter (TOM), C, K, $K_2O$, Mg, MgO, N, P, $P_2O_5$, B, Cu, Ni and Zn of the compost obtained were determined. A high quality compost was obtained in which only the EC values are slightly elevated. Particularly good was the compost obtained with *H. illucens* and *Eublaberus sp*. This quality is in agreement with previous research on the mass balance of composting. Therefore, both species offer, in the field of circular economy, encouraging prospects for the development of composting enterprises.

## Introduction

The invasion of the southern coast of Spain by the brown alga *Rugulopteryx okamurae* is unprecedented in Europe, as there has never before been a biological invasion with such intensity in such a short time [1]. *R. okamurae* belongs to the *Dictyotaceae* family and is native to the temperate waters of the northwestern Pacific (China, Taiwan, Philippines, Japan and Korea). Its first records in Europe date back to 2002 [2] being most likely introduced by the

**Data Availability Statement:** All relevant data are within the paper and its Supporting Information files.

**Funding:** All the financial support has been received by professor Jose Carlos García-Gómez and any funder have influence in the research. The details are: - JCGG (68/83 / 4081/0171) Organization of American States (https://www.oas.org/en/). - JCGG (68/83 / 4358/0171) Autoridad Portuaria de Sevilla (https://www.puertodesevilla.com/). - JCGG (68/83 / 3850/0171) Diputación Provincial de Cádiz (https://www.dipucadiz.es/). - JCGG (PRJ201903535) Fundación CEPSA (https://www.cepsa.com/es/). - JCGG (68/83 / 3608/0171) Fundación ENDESA (https://www.endesa.com/es). - JCGG (PRJ201903717) Red Eléctrica de España (https://www.ree.es/es). The funders had no role in study design, data collection and analysis, decision to publish, or preparation of the manuscript.

**Competing interests:** The authors declare that they have no known competing financial interests or personal relationships that could have appeared to influence the work reported in this paper. In fact, the technological solutions implemented here have been produced exclusively under research guidelines. The authors do not have any shareholding in any composting company, nor do they charge any subsidies for the removal of the invasive algae. There are also no political or ideological interests that could affect the research.

importation of Japanese oysters [3]. The species preferentially expands in the bottoms of the subtidal zone between 5 and 30 m, reaching up to 40 m and displacing local algal species [4]. However, it was not until 2015 when it was detected in the area of the Strait of Gibraltar [1, 5]. That same year, 5000 mt of algae were extracted from the beaches of Ceuta (North Africa) [6]. Also in 2019, the species was detected for the first time in the Azores Islands (Portugal) [7]. In 2020, the invasion occupied more than three million $m^2$ in the Strait Natural Park [1]. In the same year, only in the municipality of Tarifa 400 mt were extracted [8]. Therefore, we can affirm that *R. okamurae* is expanding without control, which has led to its inclusion in the catalog of invasive alien species (IAS) by the Ministry for Ecological Transition [9].

Among the multiple impacts of this invasive species is the enormous volume of its outcrops. These produce serious economic damage, as they reduce beach tourism along much of the coastline, as well as underwater activities [4]. The effects are also devastating in the displacement of many native species; in the alteration of the marine environment at the regional level; in changes in the productivity and biodiversity of marine ecosystems; as well as in the economic sphere on fishing and associated industries [10]. Serious effects of invasion have also been observed on the marine benthos; the overall biodiversity of large areas; the viability of many native species; and nutrient recycling [1]. Recent studies indicate that the environmental impacts are much greater on benthic communities than previous invasions of other algal species [1, 11]. In addition, some research shows that this species is able to maintain its functionality even in deep waters being able to spread its propagules many kilometers away thanks to marine currents [12]. Much work has been done to understand what biological mechanisms explain this invasion and especially why native herbivorous species do not consume this alga [13]. The main cause of this phenomenon seems to lie in the toxicity of *R. okamurae* and particularly in its high concentration of specific diterpenes (dilkamural) that are not found in native algae [14]. In summary, the high invasive capacity of *R. okamurae*, like other seaweeds, is the sum of a high capacity to spread, increased growth due to rising temperatures due to climate change, increased nutrients in the seas and its high toxicity [15]. In addition to this intrinsic capacity of algae, many marine ecosystems are highly invadable because they are highly disturbed by decades of severe impacts; present "empty niches"; have high nutrient fluctuations and greatly impaired symbiotic relationships among native species [15].

Therefore, given the severe impact of an unprecedented biological invasion and the current impotence of being able to mitigate it on the seabed, it is necessary to promote lines of rational and sustainable exploitation of the large biomass dumped by the sea on the coast (arribazones), within the scope of circular economy, since the economic cost of its removal is very high for the affected municipalities [16]. In this sense, the potential of *R. okamurae* to produce bioactive compounds (sulfated polysaccharides, polyphenols, pigments and carotenoids) applicable to cancer treatment, blood pressure problems, hyperglycemia, as antiviral, anti-inflammatory, in the reinforcement of the immune system or as neuroprotective compounds has been explored [17]. This alga has also been used in animal feed [13], for the manufacture of bioplastics [8] or in anaerobic fermentation for the production of biogas or digestates that are derived to fertilizers [18].

Another of the alternatives proposed to alleviate the costs of removing the massive *Rugulopteryx* coastal deposits of southern coast of Spain has been composting. Composting is the aerobic biodegradative process of organic residues [19]. Among the advantages of composting are the considerable reduction of waste volume [20]; its bioremediation [21]; the reduction of greenhouse gas (GHG) emissions [22, 23]; the production of protein meals [24]; biodiesel [25] and the production of organic fertilizers useful in agriculture at a time of rising prices of chemical fertilizers [7, 26].

The volume reduction of composting is undoubtedly a great advantage in many massive wastes, such as algae blooms. In this regard, previous research indicates that windrow systems with frequent turning can reduce waste volume by up to 83% [27]. Also [20], using vermicomposting, manage to decrease the biomass of algal residues combined with manure by more than 80%. In other cases, a decrease of 39% by weight and 37% by volume has been achieved with various types of waste using composting with microorganisms [28]. Another alternative tested to reduce the volume of waste is composting with invertebrates, which, being poikilotherms and needing to regulate their temperature with that of the environment, are very efficient in the transformation of waste if we provide them with the appropriate environmental conditions [29]. In fact, it is predicted that by 2030 humanity will generate more than two trillion tons of waste [30]. Couple that with the increasing volume of seaweed coastal wastes, the prospect is frightening [15]. In addition, an important advantage of composting with invertebrates is that we do not lose nitrogen as with traditional composting methods [19].

Composting in its various forms is also an interesting bioremediation procedure [31]. Marine algae have a great capacity to capture heavy metals from the environment [32], counteract eutrophication [33] or absorb greenhouse gases (GHG) [25]. In the latter case, for *R. okamurae*, some studies show significant differences between composting methodologies with and without algae [21, 22], even with fresh and desalted biomass [21]. These differences in composting results between studies seem to be due to a complex set of factors such as the nature of the waste, the level of oxygenation of compost pile, its humidity and temperature or the addition of additives such as zeolite, sepiolite, sawdust or biochar [23].

We know that since the 18th century $CO_2$ emissions have increased by 30% and $CH_4$ emissions by 50%, which is an unprecedented process in terms of intensity and speed in the history of our planet [34]. Composting has been proposed as a mechanism to mitigate these emissions [25]. In this regard, the averages in $CO_2$ equivalents of different composting systems are low compared to the cost of chemical fertilizer manufacturing. $CH_4$ and $N_2O$ emissions have an effect on $CO_2$ equivalents ($CO_2$e) 28 and 265 greater than $CO_2$ emissions [35]. It is evident that composting systems represent a clear saving of GHG emissions and recent studies indicate that with insects the emission savings would be even greater with respect to vermiculture [22]. In fact, we can consider that composting with insects and chickens hardly emits any GHG [36]. Therefore, if we combine the ability of algae to fix $CO_2$ with the ability of invertebrate composting to save emissions, we have an excellent means to considerably reduce the GHG problem in the biosphere. If we add to this the invasive character of *R. okamurae* algae, its potential to colonize spaces and generate biomass per unit of time, we contribute to alleviate several environmental problems simultaneously. In this way, we offer an economically viable alternative so that the coastal municipalities of the affected marine areas can face the high costs of removing the outcrops.

In the present work, we analyzed the composition of the compost generated by various invertebrate species fed with algae at 50% and without algae. These analyses are key to determine not only the quality of the compost generated, but also the use we can make of it if we design large-scale composting systems that are sustainable over time, environmentally and economically.

## Materials and methods

During the summer of 2022, *R. okamurae* seaweed samples were collected from beach outcrops (Fig 1) and stored in coolers at 5˚C until use for invertebrate feeding. As it is an invasive species and in public spaces, no special permits were required for the removal of the algae, although the local and regional authorities were informed of our activities. With the material

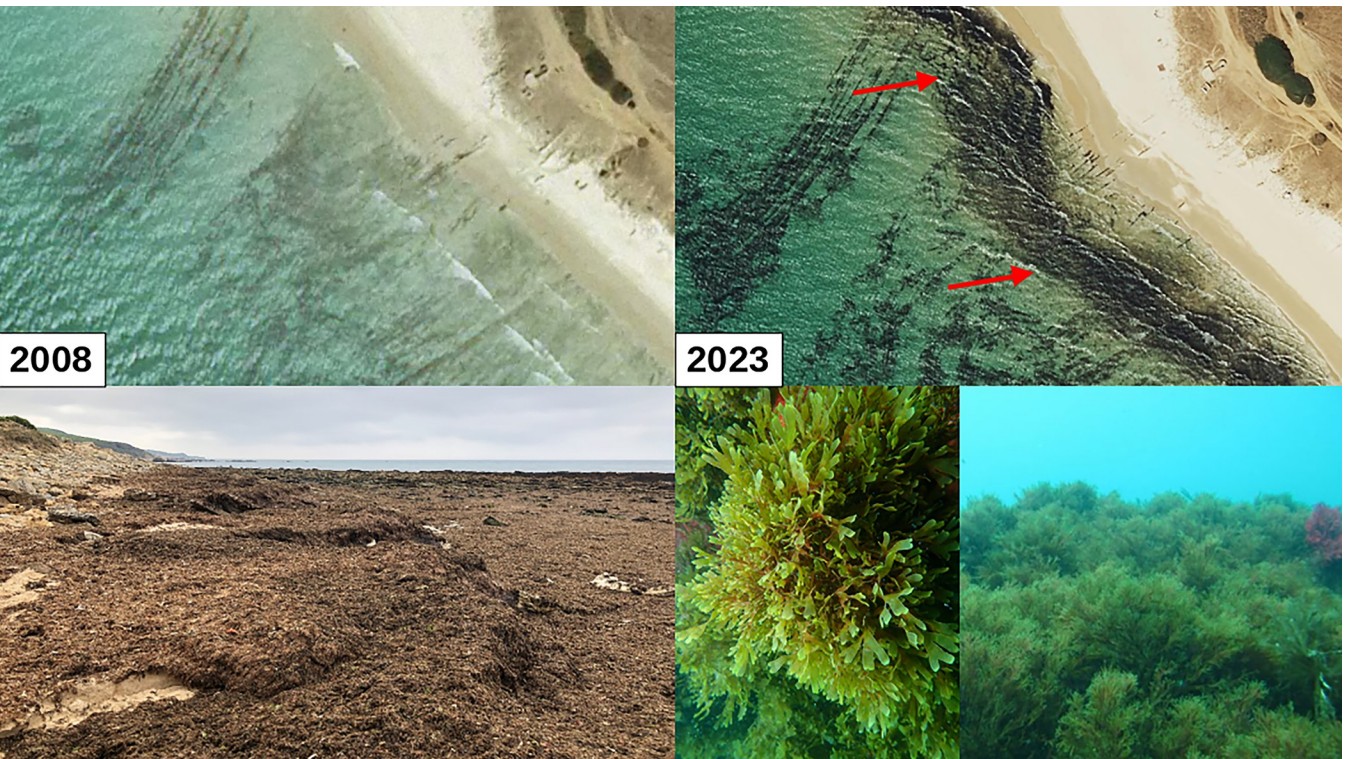

**Fig 1. Biomass deposits of the invasive alga *Rugulopteryx okamurae* in coastal areas of Tarifa, Spain.** The effect on the coastline before and after the invasion and how it covers the entire ocean floor is shown. Bottom: center and right *R. okamurae* under the sea surface. According to Google Earth's copyright guidelines the images can be used for scientific publications if they are not modified.

removed from the beaches, two sets of diets were prepared: one with 100% miscellaneous food waste (FW) (control group) and other with 50% seaweed (RO) and 50% FW (treatment group). Pure algae were not used, as previous studies showed that their toxicity can only be tolerated by a few invertebrate species [21]. By decreasing this toxicity to 50% in the experimental diet, four different invertebrate species could be used: the earthworm *Dendrobaena veneta*, black soldier fly larvaes (BSFL, *Hermetia illucens*), adults and nymphs of the cockroach *Eublaberus sp*. Ivory and mealworms (*Tenebrio molitor*).

During three months (March to May), each feed component of the organic waste diet was weighed and its percentage on the total weight was calculated. In both types of diet (with and without algae), the feed was cut into 1 cm pieces with scissors and homogenized as a mass with an industrial grinder. Feeding was *ad libitum* and dog feed was added twice a week to make up for possible protein or mineral deficits. The unconsumed remains of the previous feeding were then removed to avoid the appearance of mites or mold. Earthworms were kept in Hungry-bin continuous flow vermicomposters (https://www.hungrybin.co.nz/), cockroaches in 80x60 cm stacked plastic boxes of the Auer brand (https://www.auer-packaging.com/), mealworms in a rack system and BSFLs in 90x45x90 cm Exoterra terrariums (http://www.exo-terra.com) (Fig 2).

In the case of BSFLs, artificial light periods of 12 hours were maintained with 12000 lumens blue-white LED bulbs. This type of illumination is the most recommended to promote reproduction of this species, which also received occasional natural light [37]. All species were maintained in appropriate thermal and humidity ranges, which was not any difficult since the study was conducted during spring (see Supplementary Materials).

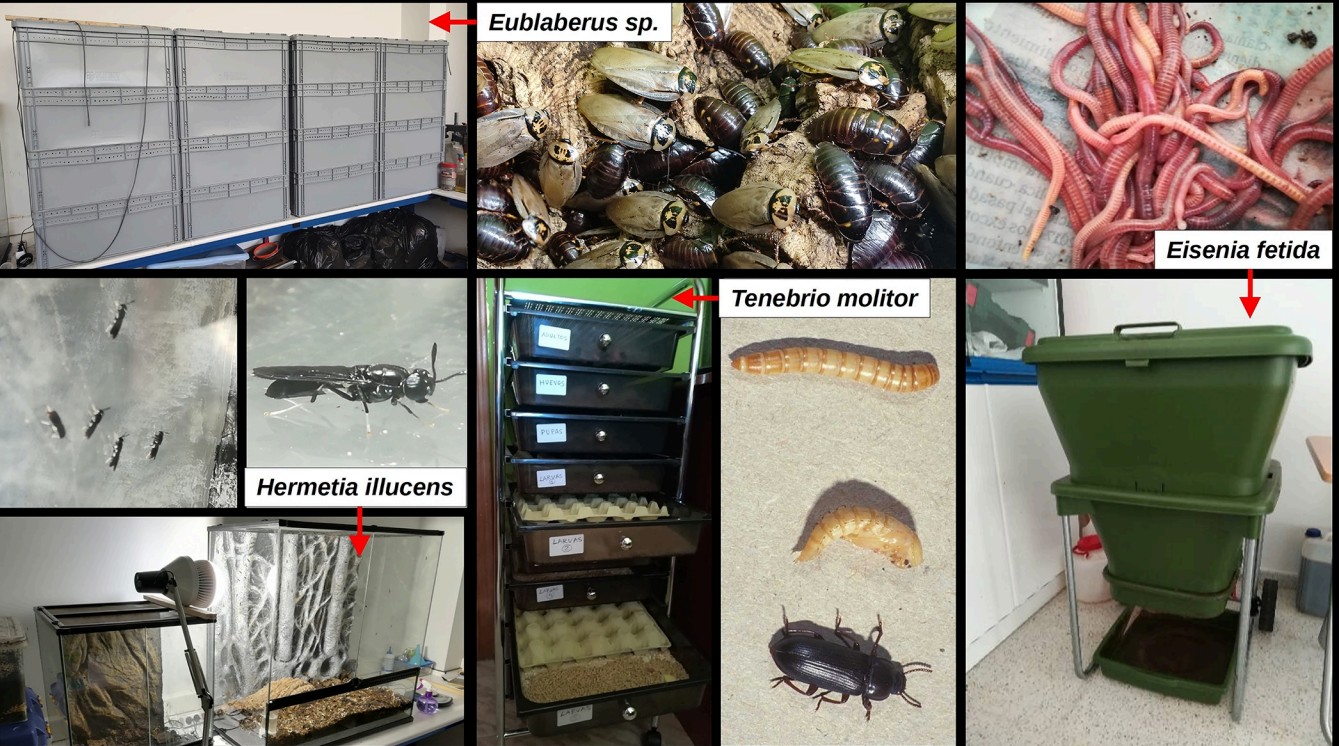

**Fig 2. Laboratory rearing systems for the invertebrate species tested in this study.**

After the three-month experimental period, five compost samples by species and treatment were extracted for a total of 40 samples (5x4x2). This is a sufficient number of replicates to determine possible significant differences using non-parametric statistical tests. The compost samples were sent to the approved laboratory from Sevilla University where the parameters Total Organic Matter (TOM, %), pH, Cation Exchange Capacity (CEC, mS/cm) as a salinity descriptor, K(%), $K_2O$ (%), Mg (%), MgO (%), P (%), $P_2O_5$ (%), C(%), N(%), C/N ratio, B(mg/kg), Zn(mg/kg), Cu(mg/kg), Ni(mg/kg) and Cd(mg/kg) were determined. This last element did not appear in any sample at detectable levels, so it was not included in the results. The parameters were analyzed in both diet groups (100% FW and 50% FW-50% RO) using the Mann-Whitney (MV) U-test [38].

To determine the quality of compost, European directives regulating the appropriate levels of pH, TOM and heavy metals were used, as well as various publications that determine the relationship between some chemical parameters and quality indexes [39]. Based on the different sources consulted, three quality levels were established for compost: A, for agricultural use including fruit trees; B, for non-agricultural use but as edaphic amendment, use in parks, gardens or in forest plantations; and C, only for use as mulch (see Supplementary Materials). Based on the above criteria, each compost sample was scored for each parameter as: unfit value (0); category C (1), category B (2) and category A (3). The final quality of the compost was the weighted average of all the score values for each parameter expressed as a percentage and assuming that the maximum quality would be 100%, i.e., a compost valid in all its parameters for agricultural use. This is equivalent to a mean index of 3. Finally, the quality indices for each diet group and for the four invertebrate species were analyzed using Dunn's nonparametric *post hoc* test of variance [40].

## Results

Fiber-rich materials such as shredded paper and coffee grounds were dominant over nitrogen-rich materials (fruits, vegetables, manure and pre-composted material) in the food waste component of the diet of animals. This allowed maintaining a C/N ratio in the ideal range between 20 and 30 (see Supplementary Materials) recommended by many studies for the material to be composted [39]. Although this ratio refers to composting based on free-living microorganisms, it seemed important to us to keep it as a reference when comparing our results with other techniques.

The values obtained (means ± standard deviations) for the chemical parameters of the two types of feed sources supplied (FW and RO) were determined (Table 1). In proportion to the residues, the *R. okamurae* algae (RO) shows about half the ideal C/N ratio, i.e. twice the N content of that recommended in traditional composting. We have already mentioned that we want to stick to this ratio even if we use invertebrates in order to be able to compare with composting techniques based on free-living microorganisms. Therefore, to ensure that the C/N ratio was also ideal in the invertebrates fed with the seaweed, we used a 50% ratio of both components (FW and RO). This gave us a C/N ratio of 20.6. Unlike the results of previous works, where RO was used as the majority component, with occasional contributions of FW to hydrate the animals, no massive mortalities were observed when both components (FW and RO) were mixed at 50% [21]. Both types of components (FW and RO) show no significant differences in pH, N, Cu or Zn. On the contrary, significant differences were observed in organic matter content (TOM), B, K, Mg and Ni with higher values in RO. In fact, B values are up to 6.8 times higher in RO. C and P values are significantly lower in RO.

Table 2 shows the chemical parameter values (means ± standard deviations) of the compost produced by black soldier fly larvae (BSFL) in the two feeding types established (100% FW and 50% FW-50% RO). The compost did not reflect significant effects between diets in the parameters TOM, Mg, MgO and B. There were significant differences in the rest of the chemical parameters. Feeding 50% RO increased the salinity of the compost as expected. The C/N ratio of the resulting compost is low for both types of feeding, but the use of RO improves it a little.

**Table 1. Chemical parameters (means ± standard deviations) of the two food sources supplied: Organic waste (FW) and *Rugulopteryx okamurae* (RO).** Mann-Whitney (MW) test statistical differences are shown.

| Parameter | FW | RO | *MW test* |
|---|---|---|---|
| CN Ratio | 27.07±11.18 | 14.11±1.50 | Z = 4.68 *** |
| pH | 6.92±0.60 | 7.94±0.16 | Z = -0.44 ns |
| Total Organic Matter (TOM) (%) | 53.00±0.71 | 63.90±2.91 | Z = -1.98 * |
| C (%) | 42.39±4.16 | 33.15±1.12 | Z = 2.03 * |
| K (%) | 1.78±1.03 | 8.05±0.37 | Z = -5.32 *** |
| Mg (%) | 0.40±0.20 | 0.79±0.11 | Z = -4.31 *** |
| N (%) | 1.57±0.37 | 2.37±0.22 | Z = -0.22 ns |
| P (%) | 0.51±0.37 | 0.15±0.01 | Z = 2.94 ** |
| B (mg/kg) | 2.84±0.93 | 192.01±6.46 | Z = -5.84 *** |
| Cu (mg/kg) | 33.11±19.09 | 7.45±1.94 | Z = -0.06 ns |
| Ni (mg/kg) | 3.41±3.05 | 5.28±0.76 | Z = -3.14 ** |
| Zn (mg/kg) | 91.89±48.13 | 19.86±1.26 | Z = -1.49 ns |

*: p-value < 0.05

**: p-value < 0.01

***: p-value < 0.001; ns = non-significant differences.

**Table 2. Chemical parameters (means ± standard deviations) of compost produced by black soldier fly larvae (BSFL) in both feeding types.** FW: Food waste. RO: *Rugulopteryx okamurae*. Mann-Whitney (MW) test statistical differences are shown.

| Parameter | BSFL (100%FW) | BSFL (50%FW—50%RO) | MW test |
|---|---|---|---|
| Electric Conductivity (EC) (mS/cm) | 4.99±0.13 | 6.57±0.55 | Z = -2.80 ** |
| CN Ratio | 8.92±0.06 | 10.94±0.22 | Z = -2.80 ** |
| pH | 6.55±0.04 | 6.85±0.12 | Z = -2.80 ** |
| Total Organic Matter (TOM) (%) | 46.80±0.24 | 47.29±4.75 | Z = -0.88 ns |
| C (%) | 24.97±0.35 | 23.85±0.75 | Z = 2.16 * |
| K (%) | 1.10±0.03 | 1.53±0.10 | Z = -2.80 ** |
| $K_2O$ (%) | 1.32±0.03 | 1.84±0.12 | Z = -2.80 ** |
| Mg (%) | 0.61±0.05 | 0.63±0.08 | Z = -0.24 ns |
| MgO (%) | 1.01±0.08 | 1.04±0.13 | Z = -0.24 ns |
| N (%) | 2.80±0.04 | 2.18±0.11 | Z = 2.80 ** |
| P (%) | 0.67±0.02 | 0.44±0.01 | Z = 2.80 ** |
| $P_2O_5$ (%) | 1.54±0.03 | 1.01±0.03 | Z = 2.80 ** |
| B (mg/kg) | 100.36±4.67 | 91.15±14.26 | Z = 0.88 ns |
| Cu (mg/kg) | 45.61±2.80 | 40.92±1.52 | Z = 2.80 ** |
| Ni (mg/kg) | 11.17±1.56 | 5.06±0.18 | Z = 2.80 ** |
| Zn (mg/kg) | 159.84±4.55 | 115.79±2.43 | Z = 2.80 ** |

\*: p-value < 0.05

\*\*: p-value < 0.01

\*\*\*: p-value < 0.001; ns = non-significant differences.

Regarding pH both types of feeding give values close to neutrality, but better in the group in which 50% of RO is added. The C, N, P and $P_2O_5$ contents decrease significantly, but so do the heavy metal contents (Cu, Ni and Zn), which indicates a certain detoxifying capacity of BSFL. According to European directives and other sources consulted [39], BSFL compost would be slightly acidic and somewhat saline for both types of diet.

Table 3 compiles the results of the chemical parameters (means ± standard deviations) of the compost produced by *Eublaberus sp.* cockroaches (blatticompost) in the two types of feeding considered. No significant differences were observed in the CN, pH, TOM, N, Cu, Ni or Zn ratios for both feeding types. Differences were observed in salinity (EC), which was again higher in the 50% RO feed, as expected. Significant increases in C, K, K2O, P and $P_2O_5$ values are also observed for the blatticompost with 50% seaweed in the feed. The increases in B in the blatticompost produced by animals fed 50% algae stand out above the non-significance of heavy metals. This pattern is exactly the opposite of that observed in BSFL indicating that the heavy metal absorption capacity of cockroaches is superior. According to the quality guidelines consulted, the blatticompost would only have as a problem its slight salinity, which in any case is lower than that observed for BSFL. Again, cockroaches are superior in their ability to detoxify the excess in minerals from food.

Table 4 compiles the results of the chemical parameters (means ± standard deviations) of compost produced by mealworms (*Tenebrio molitor*) under the two considered feeding types (FW and RO). There are significant differences in all parameters except pH due to the feeding type. Once again, the 50%RO-50%FW diet produces higher salinity (EC), K, $K_2O_5$, Mg, MgO, N, P, $P_2O_5$, B, Cu, Ni, and Zn levels, as expected. Conversely, the levels of C and TOM decrease in the algae diet. Although the mealworm frass remains slightly elevated in salinity (less than that of BSFL and blatticompost), its acidity makes it impractical for large-scale agricultural use.

**Table 3. Chemical parameters (means ± standard deviations) of compost produced by cockroaches (*Eublaberus sp.*) under both feeding types.** FW: Food waste. RO: *Rugulopteryx okamurae*. Statistical differences are shown using the Mann-Whitney test (MW).

| Parameter | Cockroaches (100%FW) | Cockroaches (50%FW-50%RO) | MW test |
|---|---|---|---|
| Electric Conductivity (EC) (mS/cm) | 4.92±1.61 | 6.24±1.28 | Z = -2.03 * |
| CN Ratio | 10.72±2.16 | 10.86±1.92 | Z = -0.39 ns |
| pH | 7.23±0.40 | 6.99±0.35 | Z = 1.31 ns |
| Total Organic Matter (TOM) (%) | 67.31±6.80 | 70.93±3.33 | Z = -1.78 ns |
| C (%) | 34.46±3.42 | 36.33±1.08 | Z = -1.88 * |
| K (%) | 1.54±0.28 | 2.24±0.25 | Z = -3.99 *** |
| $K_2O$ (%) | 1.86±0.34 | 2.70±0.30 | Z = -3.99 *** |
| Mg (%) | 0.39±0.09 | 0.63±0.15 | Z = -3.55 *** |
| MgO (%) | 0.65±0.16 | 1.05±0.25 | Z = -3.55 *** |
| N (%) | 3.32±0.63 | 3.42±0.48 | Z = -0.03 ns |
| P (%) | 0.44±0.04 | 0.54±0.09 | Z = -2.03 * |
| $P_2O_5$ (%) | 1.01±0.08 | 1.22±0.21 | Z = -2.21 * |
| B (mg/kg) | 53.73±13.63 | 66.41±14.84 | Z = -2.03 * |
| Cu (mg/kg) | 29.20±10.76 | 29.47±4.21 | Z = -0.54 ns |
| Ni (mg/kg) | 6.03±3.05 | 5.52±1.02 | Z = -0.28 ns |
| Zn (mg/kg) | 84.62±16.02 | 89.03±11.80 | Z = -1.11 ns |

*: p-value < 0.05

**: p-value < 0.01

***: p-value < 0.001; ns = non-significant differences.

**Table 4. Chemical parameters (means ± standard deviations) of compost produced by mealworms (*Tenebrio molitor*) under both feeding types.** FW: Food waste. RO: *Rugulopteryx okamurae*. Statistical differences are shown using the Mann-Whitney test (MW).

| Parameter | Mealworms (100% FW) | Mealworms (50%FW-50%RO) | MW test |
|---|---|---|---|
| Electric Conductivity (EC) (mS/cm) | 3.84±1.10 | 6.10±0.41 | Z = -2.80 ** |
| CN Ratio | 15.53±0.14 | 6.79±0.17 | Z = 2.80 ** |
| pH | 5.84±0.08 | 5.87±0.03 | Z = -0.56 ns |
| Total Organic Matter (TOM) (%) | 86.32±7.40 | 70.49±0.61 | Z = 2.80 ** |
| C (%) | 39.91±1.40 | 33.97±0.42 | Z = 2.80 ** |
| K (%) | 0.50±0.10 | 1.59±0.11 | Z = -2.80 ** |
| $K_2O$ (%) | 0.60±0.12 | 1.92±0.13 | Z = -2.80 ** |
| Mg (%) | 0.19±0.09 | 0.49±0.08 | Z = -2.80 ** |
| MgO (%) | 0.32±0.14 | 0.81±0.13 | Z = -2.80 ** |
| N (%) | 2.57±0.07 | 5.00±0.11 | Z = -2.80 ** |
| P (%) | 0.44±0.05 | 0.99±0.07 | Z = -2.80 ** |
| $P_2O_5$ (%) | 1.00±0.12 | 2.26±0.15 | Z = -2.80 ** |
| B (mg/kg) | 12.05±0.95 | 24.52±1.28 | Z = -2.80 ** |
| Cu (mg/kg) | 15.48±7.49 | 161.82±15.95 | Z = -2.80 ** |
| Ni (mg/kg) | 3.71±3.12 | 12.47±0.55 | Z = -2.80 ** |
| Zn (mg/kg) | 78.02±21.11 | 160.63±15.95 | Z = -2.80 ** |

*: p-value < 0.05

**: p-value < 0.01

***: p-value < 0.001; ns = non-significant differences.

**Table 5. Chemical parameters (means ± standard deviations) of compost produced by worms (*Dendrobaena veneta*) under both types of feeding.** FW: Food waste. RO: *Rugulopteryx okamurae*. Statistical differences are shown using the Mann-Whitney test (MW).

| Parameter | Redworms (100%FW) | Redworms (50%FW-50%RO) | MW test |
|---|---|---|---|
| Electric Conductivity (EC) (mS/cm) | 6.79±0.17 | 7.24±0.92 | Z = -0.56 ns |
| CN Ratio | 7.98±0.04 | 8.53±1.28 | Z = -0.08 ns |
| pH | 7.67±0.13 | 6.90±0.22 | Z = 2.80 ** |
| Total Organic Matter (TOM) (%) | 54.95±0.51 | 52.96±12.18 | Z = 0.08 ns |
| C (%) | 28.44±2.24 | 26.77±5.30 | Z = 0.08 ns |
| K (%) | 0.66±0.29 | 1.14±0.22 | Z = -2.48 * |
| $K_2O$ (%) | 0.79±0.35 | 1.37±0.27 | Z = -2.48 * |
| Mg (%) | 0.34±0.16 | 0.48±0.14 | Z = -1.20 ns |
| MgO (%) | 0.57±0.26 | 0.80±0.24 | Z = -1.20 ns |
| N (%) | 3.57±0.30 | 3.28±1.11 | Z = 0.08 ns |
| P (%) | 0.27±0.01 | 0.31±0.03 | Z = -2.00 * |
| $P_2O_5$ (%) | 0.63±0.01 | 0.71±0.07 | Z = -2.16 * |
| B (mg/kg) | 52.65±4.80 | 129.99±45.01 | Z = -2.80 ** |
| Cu (mg/kg) | 40.65±1.41 | 45.19±4.27 | Z = -1.84 ns |
| Ni (mg/kg) | 1.71±0.12 | 8.87±1.56 | Z = -2.80 ** |
| Zn (mg/kg) | 77.16±1.52 | 86.58±18.19 | Z = -0.08 ns |

*: p-value < 0.05

**: p-value < 0.01

***: p-value < 0.001; ns = non-significant differences.

Table 5 includes the results of the chemical parameters (means ± standard deviations) of vermicompost produced by earthworms (*Dendrobaena veneta*) under the two considered feeding types. There were no significant differences in salinity (EC), C/N ratio, TOM, C, Mg, MgO, N, Cu, and Zn due to feeding type (**Table 5**). However, significant differences were observed in pH, which was more acidic in the vermicompost derived from the RO and FW diet. The values of K, K2O, P, P2O5, B, and Ni were also higher under this feeding type (**Table 5**). The salinity of the algae compost is the highest, although its pH is appropriate (**Table 5**). This is consistent with previous studies that demonstrate the limited ability of earthworms to process the studied algae [21].

Finally, the weighting of the parameters for each sample based on the quality levels (see Supplementary Materials) yielded very clear values according to the Dunn test. The highest qualities (>70%) are obtained with BSFL fed with RO, as well as with *Eublaberus sp*. cockroaches and worms fed with FW (Fig 3). The remaining treatments are very similar, with qualities between 60% and 70%, and only in mealworms fed with algae would we obtain fertilizer qualities below 60%.

## Discussion

Coastal biomass deposits of *R. okamurae* algae are of such magnitude that they necessitate the design of strategies for their reuse, with composting emerging as one of the most promising options [21]. Previous studies have shown that invertebrate composting offers numerous advantages over composting with free-living microorganisms. One notable advantage is its ability to provide superior quality, as invertebrates excel in adjusting pH, absorbing heavy metals, and detoxifying pathogenic microorganisms [22]. It has been observed that worms feed on the bacterial soup of partially degraded materials, thereby modifying the microbiological

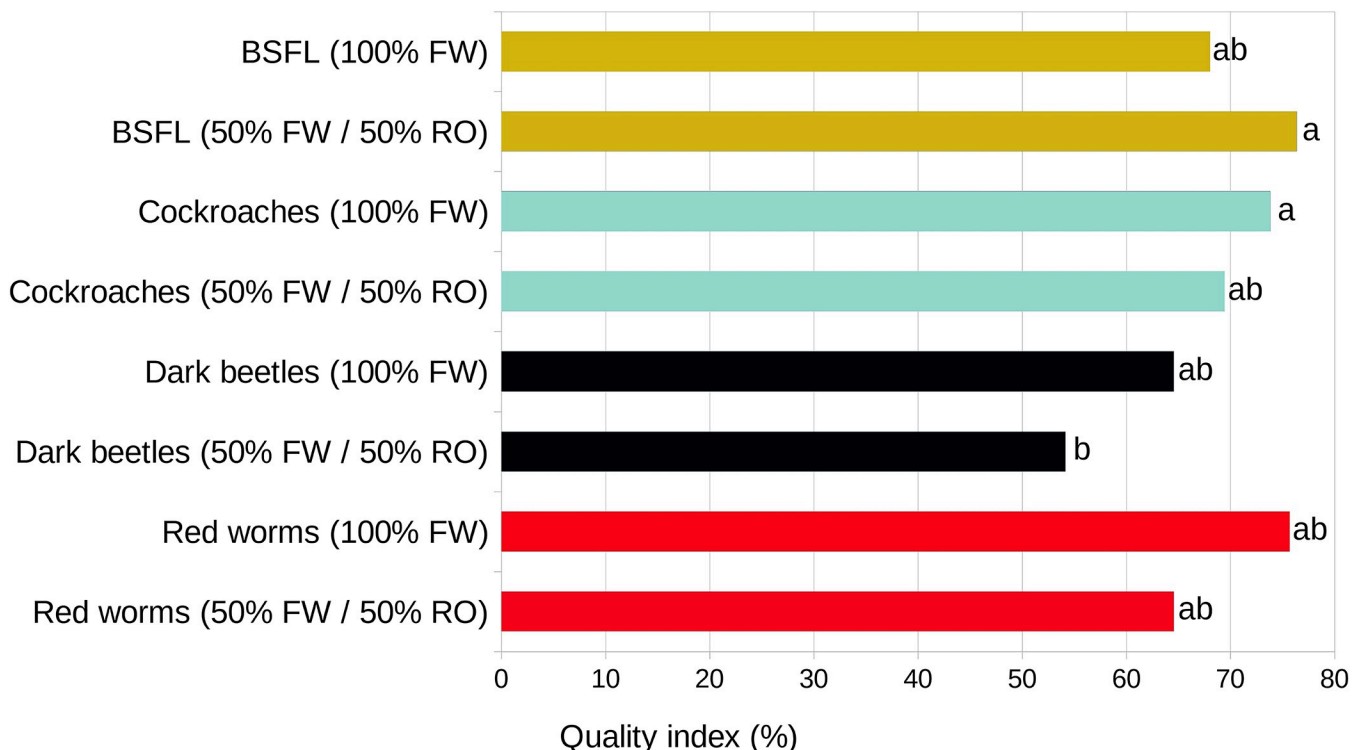

**Fig 3. Quality indices of organic fertilizer obtained with black soldier fly larvae (BSFL, *Hermetia illucens*), *Eublaberus sp.* cockroaches, mealworms (*Tenebrio molitor*) and worms (*Dendrobaena veneta*).** The letters indicate the groups obtained through the *post-hoc* non-parametric variance test of Dunn.

composition of the substrate [41]. Similar phenomena occur in insect guano [42]. Moreover, invertebrate composting is faster and more cost-effective [22]. Consequently, we selected five species of invertebrates for the experiments, as any composting solution aimed at addressing the algae accumulation should yield a product of sufficient quality to render the collection economically viable, considering the substantial costs involved for municipalities in the southern coastal area of Spain.

The *D. veneta* worms were selected over other species used in vermicomposting such as *Eisenia* (*E. andrei* and *E. fetida*) due to their greater resistance to consuming the algae, as shown in previous studies [21]. It should be noted that this species has a thicker cuticle, making it more resistant to the slightly salty environment caused by the *R. okamurae* algae during vermicomposting. Additionally, *D. veneta* is better able to withstand certain changes in pH and temperature, making it a more robust species overall [43]. Regarding temperature, our experience indicates that breeding this species is highly problematic above 28˚C, as it leads to massive die-offs. In the southern region of Spain, this species suffers from high summer temperatures, so breeding must be done in large vermicomposters that are shaded and allow for burial at a certain depth. Furthermore, these vermicomposters should be regularly irrigated not only to maintain moisture but also to cool down the substrate. Another possibility with this worm species is to exploit it in basements that maintain lower temperatures. One advantage of *D. veneta* over *Eisenia* species is its dual use in composting and as fishing bait, which adds value to its use in bioremediation [43]. However, the compost generated by this worm species tends to have slightly higher salinity levels compared to other invertebrates. This indicates that the species has a lower capacity to metabolize salts. Therefore, it does not appear to be the ideal species for composting the algae accumulations.

*Eublaberus sp.* Ivory cockroaches have shown good consumption rates of the algae and resistance to its phytotoxins in previous research [21]. However, they need temperatures above 20˚C, so their winter maintenance only requires heating occasionally in southern Spain. In contrast, the species develops correctly even at summer temperatures of 35-40˚C if it has hydration as we have been able to verify. An undoubted advantage of this species is that in addition to plant food (fruit and vegetable scraps) it can consume meat scraps, used oil, prefabricated food scraps, bread, agricultural by-products, dairy waste, paper, cardboard and even manure [22]. In other words, their diet is broader than that of earthworms, adapting to an enormous variety of components, which would make it possible to maintain the breeding facilities at full capacity all year round and not only during the period when the coastal biomass deposits occurs. Another point in favor of this species is its detoxification capacity, since the compost they generate has less heavy metals than that of mealworms or earthworms. Some studies carried out with cockroaches of the genus *Eublaberus* indicate that the composition of their intestinal microbiota is much more varied than in other insects, which is why they also detoxify the substrate microbiologically [44]. In fact, in nature they feed on bat guano, there being few habitats more dangerous than this at the microbiological level, since moisture and nitrogen-rich debris are the ideal culture medium for pathogens [45]. The compost quality values of this species are close to 70% in the diet with algae and exceed this figure in the diet with organic residues. If we wished to obtain higher quality values, we would only have to provide in their diet components low in salts, since the EC parameter is the only one that was slightly elevated.

The larvae of the soldier fly (BSFL) *H. illucens* are widely used as a source of protein in animal [46] and even human [47] feed. This species comes from the South American tropical savanna [46] so its thermal, moisture and light requirements are high [48]. In this sense, artificial lighting has been proposed by testing quartz-iodine lamps [48] and blue-white LEDs [37]. If we were to use greenhouses, we would save costs on lighting and temperature with which the cost-effectiveness of using BSFL in bioremediation would further increase. Our results show that the compost quality of 50% algae-fed BSFL is very high (~76%) and therefore it is a highly recommended species.

Finally, mealworms have similar thermal requirements to BSFL, so ensuring a temperature of at least 24˚C is a good measure if using this species over winter. Sustained temperatures above 28˚C are not suitable for this species because they cause dehydration of the smaller larvae [49]. If these temperatures are maintained, which in summer occur easily in southern Spain, water sprays would be obligatory, which would produce an increase in mites that are a problem for this species. Therefore, the best strategy is to maintain a dry but not hot environment and to administer water in the feed by adding chopped vegetables. In addition, tenebrio larvae have shown difficulties in completing metamorphosis when supplied with *R.okamurae* algae as the sole food source due to their sensitivity to the algae's diterpenes [21]. Finally, our results indicate that the compost (guano) of this species is the lowest quality of the four tested, although it could be used in gardens or forest plantations. Therefore, we advise against the use of mealworms in algae bioremediation. However, due to the potential of this species in bioremediation we believe that we should further explore its possibilities [50].

In view of the obtained compost quality results, we can conclude that *Rugulopteryx* bioremediation is highly reliable with two insect species: BSFL and *Eublaberus* cockroaches. By increasing the low salt content components in the insects' diet, we could obtain a 100% quality fertilizer. This could be used on a large scale in a wide range of agricultural plantations. In the current context of increasing prices of chemical fertilizers, this is certainly an added value. We know that chemical fertilizers damage the soil microbiota in the long term, resulting in high GHG emission [51]. In contrast, the use of insects for composting allows us to save these GHG

emissions in multiple ways. On the one hand, we avoid the high emissions from untreated waste [52]. On the other, by incorporating organic matter into the soil and improving its structure we can dispense with plowing, so we do not release $N_2O$ into the atmosphere [53]. In addition, the high manufacturing costs and emissions associated with the processing of chemical fertilizers are reduced [51]. For all these reasons, we recommend a detailed study of the feasibility of implementing insect farms of both species in the areas with the highest incidence of *R. okamurae* algae. By doing so, we reduce extraction costs and offer a needed product for agriculture. Therefore, there is a need to facilitate this process, not only in making investments, but also in implementing effective legislative measures that will help this emerging biotechnology [54].

## Conclusions

Composting with invertebrates of the invasive alga *Rugulopteryx okamurae* is feasible when it is mixed at 50% with organic residues to reduce its toxicity. The quality of the compost is good but with certain salinity. The best results are obtained with cockroaches of the genus *Eublaberus* and black soldier fly larvae (BSFL). Our results applied on a large scale allow us to reduce the high costs of beach algal biomass removal.

## Supporting information

**S1 Data. Compost *Rugulopteryx*.**
(XLSX)

## Acknowledgments

We are grateful for the facilities provided by the Port Authority of the Bay of Algeciras (APBA) and the Port Authority of Seville and the Aquarium of Seville. We are grateful for the collaboration of Puerto Deportivo La Alcaidesa (La Línea).

## Author Contributions

**Investigation:** Daniel Patón, José Carlos García-Gómez.

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
