## [Decision Letter · Decision Letter 0]

12 Jan 2024

PONE-D-23-38768Invertebrate composting quality of the invasive alga Rugulopteryx okamurae, prospects for its bio-recycling, management and circular economyPLOS ONE

Dear Dr. Patón,

Thank you for submitting your manuscript to PLOS ONE. After careful consideration, we feel that it has merit but does not fully meet PLOS ONE’s publication criteria as it currently stands. Therefore, we invite you to submit a revised version of the manuscript that addresses the points raised during the review process.

We look forward to receiving your revised manuscript.

Kind regards,

Noé Aguilar-Rivera

Academic Editor

PLOS ONE

Journal Requirements:

All the financial support has been received by professor Jose Carlos García-Gómez and any funder have influence in the research. The details are:

- JCGG (68/83 / 4081/0171) Organization of American States (https://www.oas.org/en/).

- JCGG (68/83 / 4358/0171) Autoridad Portuaria de Sevilla (https://www.puertodesevilla.com/).

- JCGG (68/83 / 3850/0171) Diputación Provincial de Cádiz (https://www.dipucadiz.es/).

- JCGG (PRJ201903535) Fundación CEPSA (https://www.cepsa.com/es/).

- JCGG (68/83 / 3608/0171) Fundación ENDESA (https://www.endesa.com/es).

- JCGG (PRJ201903717) Red Eléctrica de España (https://www.ree.es/es).

This study was financed by the Red Eléctrica de España (REE) and CEPSA Foundation.  Additional financial collaboration was provided by Diputación de Cádiz and Puerto Deportivo La Alcaidesa (La Línea). This project also benefited from scientific equipment and infrastructures financed by the Port Authority (AP) of Seville and the Aquarium of Seville

All the financial support has been received by professor Jose Carlos García-Gómez and any funder have influence in the research. The details are:

- JCGG (68/83 / 4081/0171) Organization of American States (https://www.oas.org/en/).

- JCGG (68/83 / 4358/0171) Autoridad Portuaria de Sevilla (https://www.puertodesevilla.com/).

- JCGG (68/83 / 3850/0171) Diputación Provincial de Cádiz (https://www.dipucadiz.es/).

- JCGG (PRJ201903535) Fundación CEPSA (https://www.cepsa.com/es/).

- JCGG (68/83 / 3608/0171) Fundación ENDESA (https://www.endesa.com/es).

- JCGG (PRJ201903717) Red Eléctrica de España (https://www.ree.es/es).

6. We note that Figure 1 in your submission contain copyrighted images. All PLOS content is published under the Creative Commons Attribution License (CC BY 4.0), which means that the manuscript, images, and Supporting Information files will be freely available online, and any third party is permitted to access, download, copy, distribute, and use these materials in any way, even commercially, with proper attribution. For more information, see our copyright guidelines: http://journals.plos.org/plosone/s/licenses-and-copyright.

We require you to either present written permission from the copyright holder to publish these figures specifically under the CC BY 4.0 license, or remove the figures from your submission:

Reviewers' comments:

Reviewer's Responses to Questions

**Comments to the Author**

1. Is the manuscript technically sound, and do the data support the conclusions?

Reviewer #1: Yes

Reviewer #2: No

Reviewer #3: Partly

2. Has the statistical analysis been performed appropriately and rigorously? 

Reviewer #1: Yes

Reviewer #2: No

Reviewer #3: Yes

3. Have the authors made all data underlying the findings in their manuscript fully available?

Reviewer #1: Yes

Reviewer #2: No

Reviewer #3: Yes

4. Is the manuscript presented in an intelligible fashion and written in standard English?

Reviewer #1: Yes

Reviewer #2: No

Reviewer #3: No

5. Review Comments to the Author

Reviewer #1: The paper is well-written and adds to the existing literature on the subject. The introduction is well-composed and has been developed on the right lines. The most appropriate methodology has been used for analysis of data and information. The results have been reported well. A logical sequence of interpretations has been followed and developed scientifically. The discussion has been well brought out and the cogency of arguments are well thought out. The discussion is comprehensive and complete. References are as required. May be accepted for publication

Reviewer #2: The authors present a potentially interesting bioremediation study that seeks to solve the upwelling problem of the invasive macroalgae Rugulopteryx okamurae by using its biomass for composting. The authors show the quality and composition of the composts generated by 4 different invertebrates fed with and without the invasive macroalgae Rugulopteryx okamurae algae. The study is interesting and relevant but has many issues.My first concern is the choice of statistical tests that don't seem correct. As I understand the experiment performed has 2 different factors: 1- invertebrate species (with 4 levels), and 2 - type of food, or treatment (with 2 levels), however, the authors didn't seem to consider this, as they used a simple Mann-Whitney test to compare the 2 different treatments within the same invertebrate species. Isolating the factors from the same experiment is controversial and not at all fully accepted in the scientific community. This is because it can lead to the wrong interpretation of the data. The ideal statistical test for this data would be a PERMANOVA (if non-parametric) or 2-way ANOVA (if parametric). Similarly, the use of Dunn test after the Kruskal-Wallis (I assume it was Kruskal-Wallis, because the authors only mentioned the Dunn post-hoc test) on Fig. 3 is wrong - it should, again, be either PERMANOVA or 2-way ANOVA. The other big problem I noticed here was the lack of control for this experiment. This should be a control for both food sources without the presence of any invertebrates to account for microbial decomposition.On a minor comment, the manuscript has many grammar mistakes and misspellings.With that, I believe that the manuscript as it is, is not acceptable for publication. I do think it has potential and would suggest the authors redo the experiment including the controls, and use the proper statistical tests.

Reviewer #3: The study sounds interesting but there is no novelty its conclusion is similar to previously published studies and the manuscript is not properly structured. Figures are not good not properly numbered and not properly described in caption. please read the .doc file

6. PLOS authors have the option to publish the peer review history of their article (what does this mean?). If published, this will include your full peer review and any attached files.

Reviewer #1: No

Reviewer #2: No

Reviewer #3: No

---

## [Author Response · Author response to Decision Letter 0]

11 Mar 2024

Response to reviewers (in capitals) of manuscript PONE-D-23-38768 entitled:

Invertebrate composting quality of the invasive alga Rugulopteryx okamurae, prospects for its bio-recycling, management and circular economy

Dear Dr. Patón,

Thank you for submitting your manuscript to PLOS ONE. After careful consideration, we feel that it has merit but does not fully meet PLOS ONE’s publication criteria as it currently stands. Therefore, we invite you to submit a revised version of the manuscript that addresses the points raised during the review process.

DEAR EDITOR,

THANK YOU SO MUCH BY THIS OPPORTUNITY.

OK

OK

OK

THERE WERE INDEED SOME DISCREPANCIES THAT WE HAVE RESOLVED IN THE ACKNOWLEDGEMENTS. THERE IS NO NEED TO CHANGE ANYTHING ELSE. THANK YOU

THE LABORATORY PROTOCOLS THAT AFFECT THE MAINTENANCE OF THE ANIMALS ARE VERY SIMPLE TO UNDERSTAND AND WE DO NOT BELIEVE IT IS NECESSARY TO INCLUDE THEM IN A SEPARATE FILE. IN ADDITION, THE LABORATORY ANALYSES FOLLOW THE OFFICIAL AOAC GUIDELINES SO THE PROTOCOLS ARE WELL KNOWN. NEVERTHELESS, THANK YOU.

We look forward to receiving your revised manuscript.

Kind regards,

Noé Aguilar-Rivera

Academic Editor

PLOS ONE

THANKS

Journal Requirements:

OK

OK. WE HAVE MENTIONED IN THE TEXT THAT THE DATA WILL BE SENT UPON REASONED REQUEST. IF THE MANUSCRIPT IS FINALLY ACCEPTED THERE IS NO OBJECTION TO UPLOAD A SUMMARY OF THE DATA IN A SEPARATE FILE. HOWEVER, THE LABORATORY THAT PERFORMED THE ANALYSES BELONGS TO THE CSIC (SPANISH NATIONAL COUNCIL OF RESEARCH) AND HAS CONFIDENTIALITY AGREEMENTS WITH THE UNIVERSITY. 

ALL SAMPLE COLLECTION IS DONE IN PUBLIC PLACES AND NO PERMITS ARE REQUIRED ALTHOUGH WE INFORM LOCAL AND REGIONAL AUTHORITIES. THIS IS MENTIONED IN THE TEXT.

All the financial support has been received by professor Jose Carlos García-Gómez and any funder have influence in the research. The details are:

- JCGG (68/83 / 4081/0171) Organization of American States (https://www.oas.org/en/).

- JCGG (68/83 / 4358/0171) Autoridad Portuaria de Sevilla (https://www.puertodesevilla.com/).

- JCGG (68/83 / 3850/0171) Diputación Provincial de Cádiz (https://www.dipucadiz.es/).

- JCGG (PRJ201903535) Fundación CEPSA (https://www.cepsa.com/es/).

- JCGG (68/83 / 3608/0171) Fundación ENDESA (https://www.endesa.com/es).

- JCGG (PRJ201903717) Red Eléctrica de España (https://www.ree.es/es).

Please provide an amended statement that declares all the funding or sources of support (whether external or internal to your organization) received during this study, as detailed online in our guide for authors at http://journals.plos.org/plosone/s/submit-now. Please also include the statement “There was no additional external funding received for this study.” in your updated Funding Statement. Please include your amended Funding Statement within your cover letter. We will change the online submission form on your behalf.

THERE WAS INDEED A DISCREPANCY BETWEEN THE DOCUMENTS. WE HAVE ADDED THE MISSING INSTITUTIONS IN THE ACKNOWLEDGEMENTS.

This study was financed by the Red Eléctrica de España (REE) and CEPSA Foundation. Additional financial collaboration was provided by Diputación de Cádiz and Puerto Deportivo La Alcaidesa (La Línea). This project also benefited from scientific equipment and infrastructures financed by the Port Authority (AP) of Seville and the Aquarium of Seville

All the financial support has been received by professor Jose Carlos García-Gómez and any funder have influence in the research. The details are:

- JCGG (68/83 / 4081/0171) Organization of American States (https://www.oas.org/en/).

- JCGG (68/83 / 4358/0171) Autoridad Portuaria de Sevilla (https://www.puertodesevilla.com/).

- JCGG (68/83 / 3850/0171) Diputación Provincial de Cádiz (https://www.dipucadiz.es/).

- JCGG (PRJ201903535) Fundación CEPSA (https://www.cepsa.com/es/).

- JCGG (68/83 / 3608/0171) Fundación ENDESA (https://www.endesa.com/es).

- JCGG (PRJ201903717) Red Eléctrica de España (https://www.ree.es/es).

AS MENTIONED ABOVE, THE MISSING INSTITUTIONS HAVE BEEN ADDED TO THE ACKNOWLEDGMENTS.

6. We note that Figure 1 in your submission contain copyrighted images. All PLOS content is published under the Creative Commons Attribution License (CC BY 4.0), which means that the manuscript, images, and Supporting Information files will be freely available online, and any third party is permitted to access, download, copy, distribute, and use these materials in any way, even commercially, with proper attribution. For more information, see our copyright guidelines: http://journals.plos.org/plosone/s/licenses-and-copyright.

We require you to either present written permission from the copyright holder to publish these figures specifically under the CC BY 4.0 license, or remove the figures from your submission:

THE AERIAL IMAGES ARE COPYRIGHT FREE, FOLLOW OPEN DATA GUIDELINES AND COME FROM THE VIEWER OF THE NATIONAL GEOGRAPHIC INSTITUTE OF SPAIN (HTTPS://WWW.IGN.ES/IBERPIX/VISOR/).

Reviewers' comments:

Reviewer's Responses to Questions

Comments to the Author

1. Is the manuscript technically sound, and do the data support the conclusions?

Reviewer #1: Yes

Reviewer #2: No

Reviewer #3: Partly

2. Has the statistical analysis been performed appropriately and rigorously?

Reviewer #1: Yes

Reviewer #2: No

Reviewer #3: Yes

3. Have the authors made all data underlying the findings in their manuscript fully available?

Reviewer #1: Yes

Reviewer #2: No

Reviewer #3: Yes

THE ONLY RESTRICTION IS THE CONFIDENTIALITY AGREEMENT OF THE CSIC (SPANISH NATIONAL RESEARCH COUNCIL) LABORATORY TO USE THE CHEMICAL DATA. THIS WAS ADDED IN THE MANUSCRIPT.

4. Is the manuscript presented in an intelligible fashion and written in standard English?

Reviewer #1: Yes

Reviewer #2: No

Reviewer #3: No

5. Review Comments to the Author

Reviewer #1: The paper is well-written and adds to the existing literature on the subject. The introduction is well-composed and has been developed on the right lines. The most appropriate methodology has been used for analysis of data and information. The results have been reported well. A logical sequence of interpretations has been followed and developed scientifically. The discussion has been well brought out and the cogency of arguments are well thought out. The discussion is comprehensive and complete. References are as required. May be accepted for publication

Reviewer #2: The authors present a potentially interesting bioremediation study that seeks to solve the upwelling problem of the invasive macroalgae Rugulopteryx okamurae by using its biomass for composting. The authors show the quality and composition of the composts generated by 4 different invertebrates fed with and without the invasive macroalgae Rugulopteryx okamurae algae. The study is interesting and relevant but has many issues.My first concern is the choice of statistical tests that don't seem correct. As I understand the experiment performed has 2 different factors: 1- invertebrate species (with 4 levels), and 2 - type of food, or treatment (with 2 levels), however, the authors didn't seem to consider this, as they used a simple Mann-Whitney test to compare the 2 different treatments within the same invertebrate species. Isolating the factors from the same experiment is controversial and not at all fully accepted in the scientific community. This is because it can lead to the wrong interpretation of the data. The ideal statistical test for this data would be a PERMANOVA (if non-parametric) or 2-way ANOVA (if parametric). Similarly, the use of Dunn test after the Kruskal-Wallis (I assume it was Kruskal-Wallis, because the authors only mentioned the Dunn post-hoc test) on Fig. 3 is wrong - it should, again, be either PERMANOVA or 2-way ANOVA. The other big problem I noticed here was the lack of control for this experiment. This should be a control for both food sources without the presence of any invertebrates to account for microbial decomposition.On a minor comment, the manuscript has many grammar mistakes and misspellings.With that, I believe that the manuscript as it is, is not acceptable for publication. I do think it has potential and would suggest the authors redo the experiment including the controls, and use the proper statistical tests.

WE APPRECIATE THE REVIEWER'S SUGGESTIONS. HOWEVER, OUR OBJECTIVE IN THIS WORK WAS NOT TO EXPLORE THE INTERACTION BETWEEN VARIABLES BY MULTIVARIATE ANALYSIS, BUT ONLY TO DETERMINE TO WHAT EXTENT EACH CHEMICAL PARAMETER CHANGES SIGNIFICANTLY BETWEEN THE TWO TYPES OF DIET. WE DO NOT INTEND TO EXPLORE THE INTERACTION BETWEEN SPECIES AND DIET BECAUSE THESE ARE INDEPENDENTLY MANAGED SPECIES. THE BATCHES ARE EXPERIMENTALLY SEPARATED. WE ONLY INTEND TO FIND OUT WHICH OF THE FOUR SPECIES GIVES BETTER RESULTS IN TERMS OF GUANO QUALITY A

---

## [Decision Letter · Decision Letter 1]

9 Apr 2024

PONE-D-23-38768R1Invertebrate composting quality of the invasive alga Rugulopteryx okamurae, prospects for its bio-recycling, management and circular economyPLOS ONE

Dear Dr. Patón,

Thank you for submitting your manuscript to PLOS ONE. After careful consideration, we feel that it has merit but does not fully meet PLOS ONE’s publication criteria as it currently stands. Therefore, we invite you to submit a revised version of the manuscript that addresses the points raised during the review process.

Review Comments to the Author

Reviewer #2: The authors haven't addressed any of my concerns regarding the lack of controls for the experiments (i.e., control for both food sources without the presence of any invertebrates to account for microbial decomposition). I would appreciate the use of the correct statistical analysis, as I have mentioned before. However, the experiment should also have the proper controls.

Reviewer #4: This paper touches on an interesting topic about how to handle invasive species by using composting with different invertebrate species. The problem, strategy, and results are clearly described and the discussion is adequate as well. I have some minor comments about the paper available in te attached file.

We look forward to receiving your revised manuscript.

Kind regards,

Noé Aguilar-Rivera

Academic Editor

PLOS ONE

Additional Editor Comments :

**Comments to the Author**

1. If the authors have adequately addressed your comments raised in a previous round of review and you feel that this manuscript is now acceptable for publication, you may indicate that here to bypass the “Comments to the Author” section, enter your conflict of interest statement in the “Confidential to Editor” section, and submit your "Accept" recommendation.

Reviewer #2: (No Response)

Reviewer #4: All comments have been addressed

2. Is the manuscript technically sound, and do the data support the conclusions?

Reviewer #2: Partly

Reviewer #4: Yes

3. Has the statistical analysis been performed appropriately and rigorously? 

Reviewer #2: No

Reviewer #4: Yes

4. Have the authors made all data underlying the findings in their manuscript fully available?

Reviewer #2: Yes

Reviewer #4: Yes

5. Is the manuscript presented in an intelligible fashion and written in standard English?

Reviewer #2: No

Reviewer #4: Yes

6. Review Comments to the Author

Reviewer #2: The authors haven't addressed any of my concerns regarding the lack of controls for the experiments (i.e., control for both food sources without the presence of any invertebrates to account for microbial decomposition). I would appreciate the use of the correct statistical analysis, as I have mentioned before. However, the experiment should also have the proper controls.

Reviewer #4: This paper touches on an interesting topic about how to handle invasive species by using composting with different invertebrate species. The problem, strategy, and results are clearly described and the discussion is adequate as well. I have some minor comments about the paper available in te attached file.

7. PLOS authors have the option to publish the peer review history of their article (what does this mean?). If published, this will include your full peer review and any attached files.

Reviewer #2: No

Reviewer #4: No

---

## [Author Response · Author response to Decision Letter 1]

10 Jul 2024

Dear editors and reviewers:

Minor syntax changes have been corrected. In the text we have clarified that our control group is the one constituted by the feeding with 100% waste and the treatment group is the one included with 50% kitchen waste and 50% with seaweed. The seaweed alone does not compost easily due to its toxicity. In fact, months can pass and it does not finish composting as we have verified in our laboratory. Including composting with microorganisms would be a different job. We want to see the differences between invertebrates not between invertebrates and other methods.

Daniel Patón

---

## [Decision Letter · Decision Letter 2]

27 Aug 2024

Invertebrate composting quality of the invasive alga Rugulopteryx okamurae, prospects for its bio-recycling, management and circular economy

PONE-D-23-38768R2

Dear Dr. Daniel Patón

We’re pleased to inform you that your manuscript has been judged scientifically suitable for publication and will be formally accepted for publication once it meets all outstanding technical requirements.

Kind regards,

Noé Aguilar-Rivera

Academic Editor

PLOS ONE

---

## [Editor Report · Acceptance letter]

24 Sep 2024

PONE-D-23-38768R2 

PLOS ONE

Dear Dr. Patón, 

I'm pleased to inform you that your manuscript has been deemed suitable for publication in PLOS ONE. Congratulations! Your manuscript is now being handed over to our production team.

Kind regards, 

on behalf of

Dr. Noé Aguilar-Rivera 

Academic Editor

PLOS ONE